# Sparse Regularized Deep Neural Networks For Efficient Embedded Learning

## Abstract

Deep learning is becoming more widespread in its application due to its power in solving complex classification problems. However, deep learning models often require large memory and energy consumption, which may prevent them from being deployed effectively on embedded platforms, limiting their applications. This work addresses the problem by proposing methods *Weight Reduction Quantisation* for compressing the memory footprint of the models, including reducing the number of weights and the number of bits to store each weight. Beside, applying with sparsity-inducing regularization, our work focuses on speeding up stochastic variance reduced gradients (SVRG) optimization on non-convex problem. Our method that mini-batch SVRG with $\ell 1$ regularization on non-convex problem has faster and smoother convergence rates than SGD by using adaptive learning rates. Experimental evaluation of our approach uses MNIST and CIFAR-10 datasets on LeNet-300-100 and LeNet-5 models, showing our approach can reduce the memory requirements both in the convolutional and fully connected layers by up to $60\times$ without affecting their test accuracy.

## 1 Introduction

Artificial intelligence is finding wider application across a number of domains where computational resources can vary from large data centres to mobile devices. However, state-of-the-art techniques such as deep learning (LeCun et al., 2015) require significant resources, including large memory requirements and energy consumption. Reducing the size of the deep learning model to a compact model that has small memory footprint without compromising its performance is a desirable research aim to address the challenges for deploying these leading approaches on mobile devices. $\ell 1$ regularization can be used as a penalty to train models to prevent the model from over-fitting the training data. As well as providing, $\ell 1$ regularization is a powerful compression techniques to penalize some weights to be zero. As the results, our research focus on improving the method based on $\ell 1$ regularization to reduce memory requirements. Moreover, as deep neural network optimization is a non-convex problem, the optimization can be stuck in local-minimal, which can reduce the performance. To address the problem, we improve SGD optimization for non-convex function to enhancing sparse representations obtained with $\ell 1$ regularization. In this paper, we propose our compression method *Weight Reduction Quantisation* which reduces both the number of weights and bits-depth of model without sacrificing accuracy. To reduces the number of weights, our method employs sparsity-inducing $\ell 1$ regularization to encourage many connections in both convolutional and fully connected layers to be zero during the training process. Formally, in this paper we consider the following unconstrained minimization problem, Given training labels $y_1, y_2, ..., y_N$ as correct outputs for input data $x_1, x_2, ..., x_N$, the optimization problem to estimate the weights in all layers, $\mathbf{W}$, is defined by

$$\min_{\mathbf{W}} \frac{1}{N} \sum_{i=1}^{N} \mathcal{L}(y_i, f(x_i; \mathbf{W})) + \lambda r(\mathbf{W}), \qquad (1)$$

where $\lambda$ is a hyper-parameter controlling the degree of regularization and the weights in all layers is given by $\mathbf{W}$. The problem 1 can be strongly convex or possibly non-convex

(Allen-Zhu & Yuan, 2016). Following update rule, the mini-batch SGD method with $\ell 1$ regularization is a popular approach for performing the optimization, and the weight update rule is given by

$$w_j^{k+1} = w_j^k - \eta_k \frac{\partial}{\partial w_j} \left( \frac{1}{B} \sum_{i=1}^{B} \mathcal{L}(y_i, f(x_i; W)) + \frac{\lambda}{M} \sum_{j=1}^{M} |w_j| \right), \qquad (2)$$

where each weight of network can be represented by $w_j$, the total number of weights is $M$. $k$ is the iteration counter and $\eta_k$ is the learning rate and $B$ is mini-batch size $(1 < B < N)$ used to approximate the full gradient. However, SGD optimization with $\ell 1$ regularization has two challenges: firstly, it inefficiently encourages weight to be zero due to fluctuations generated by SGD (Tsuruoka et al., 2009). Secondly, SGD optimization slowing down convergence rate due to the high variance of gradients. The two methods of *cumulative $\ell 1$ regularization* and *SVRG* can solve the two challenges respectively:

**Cumulative $\ell 1$ regularization**  Tsuruoka et al. (2009) proposed a method cumulating the $\ell 1$ penalties to resolve the problem. The method clips regularization at zero, which avoids the derivative $\frac{\partial}{\partial w_j} \sum_{j=1}^{M} (\frac{\lambda}{M} |w_j|)$ being non-differentiable when $w_j = 0$ and provides a more stable convergence for the weights. Moreover, the cumulative penalty can reduce the weight to zero more quickly.

**Mini-batch SVRG**  As SGD optimization has slow convergence asymptotically due to noise, Johnson & Zhang (2013) proposed SVRG that can efficiently decrease the noise of SGD by reducing the variance of gradients by:

$$w_j^{k+1} = w_j^k - \eta_k \left( \frac{1}{B} \sum_{i=1}^{B} (\nabla \psi_i(w_j^k) - \nabla \psi_i(\tilde{w}_j)) + \tilde{\mu}_j \right), \qquad (3)$$

where $\tilde{\mu}_j$ is the average gradient of sub-optimal weights $\tilde{w}_j$ which is the weight after every $m$ SGD iterations

$$\begin{aligned} \tilde{\mu}_j &= \frac{1}{N} \sum_{i=1}^{N} \frac{\partial \mathcal{L}(y_i, f(x_i; \tilde{W}))}{\partial w_j} \\ &= \frac{1}{N} \sum_{i=1}^{N} \nabla \psi_i(\tilde{w}_j), \end{aligned} \qquad (4)$$

where $\tilde{W}$ is the sub-optimal weights after $m$ SGD iterations in all layers. For succinctness we also write $\nabla \psi_i(w_j^k) = \frac{\partial \mathcal{L}(y_i, f(x_i; W))}{\partial w_j}$. They determined that reduction of variance helps initial weights $w_0$ close to global minima at the beginning in order to boost the convergence rate of SGD in strongly convex problems. Johnson & Zhang (2013) further prove that the performance of SGD degrades with mini-batching by the theoretical result of complexity. Specifically, for batch size of $B$, SGD has a $1/\sqrt{B}$ dependence on the batch size. In contrast, SVRG in a parallel setting has $1/B$ dependence on the batch size which is much better than SGD. Hence, SVRG allows more efficient mini-batching. However, for non-strongly convex problems, global minimization of non-convex function is NP-hard(Allen Zhu & Hazan, 2016). Johnson & Zhang (2013) have a assumption that SVRG can also be applied in neural networks to accelerate the local convergence rate of SGD. Further, Allen Zhu & Hazan (2016) prove non-asymptotic rates of convergence of SVRG for non-convex optimization and proposed improved SVRG that is provably faster than SGD. Hence, a promising approach is to use mini-batch SVRG instead of SGD with cumulative $\ell 1$ regularization.

**Main Contributions**  We summarize our main contributions below:

1. **Reducing memory requirements**:

    1.1 We analyse a method that combines SVRG with cumulative $\ell 1$ regularization to reduce the number of weights, and propose our method *Delicate-SVRG-cumulative-$\ell 1$* which can significantly reduce the number of weights by up to $25\times$

without affecting their test accuracy. To our knowledge, ours is the first work that to combine mini-batch SVRG with cumulative $\ell 1$ regularization for non-convex optimization.

1.2 To further reduce the memory requirements of models, we aim to reduces the number of bits to store each weight. Compression method *Weight Reduction Quantisation*, including both reducing number of weights and bit-depth, can reduce the memory footprints up to $60\times$ without affecting accuracy.

2. **Accelerating convergence rates**:

2.1 We analyse non-convex stochastic variance reduced gradient (SVRG). Based on the results from (Reddi et al., 2016), we provide the condition when SVRG has faster rates of convergence than SGD.

2.2 We empirically show that modified SVRG in our method have faster rates of convergence than ordinary SVRG and SGD.

## 2 Related Works

Different methods have been proposed to remove redundancy in deep learning models. Sparse representation is a good approach to reduce the number of parameters. Han et al. mainly explored pruning which is a direct approach to remove small values of connection and focuses on the important connections with large weight values in all layers of the network. However, a disadvantage is that after pruning the needs networks to be retrained. One idea from matrix factorization can be applied to compressed parameters in models by finding a low rank approximation of the weight matrix Denton et al. (2014). However, in practice whilst it improves computation performance, it dose not significantly reduce memory requirements.

Weight sharing aims to approximate weights by a single weight. Chen et al. proposed HashedNets binning network connections into hash buckets uniformly at random by a hash function. As part of a three stage compression pipeline, Han et al. use k-means clustering to identify the shared weights for each layer of a trained network.

Weight quantization for reducing the bit-width to store each weight is an other approach to reduce memory requirements of models. Gysel et al. can successfully condense CaffeNet and SqueezeNet to 8 bits with only slight accuracy loss. Han et al. quantizes the sparse weights matrix to be an index which encodes in 8-bit for convolutional layers and 5-bit for fully connected layers. Rastegari et al. used binary operations to find the best approximations of the convolutions, in which the bit-size can be reduced to 1-bit.

Another type of approach uses regularization to induce sparsity. Hinton et al. proposed "dropout" that refers to dropping out neurons that are from visible and hidden layers in neural network during training, which can be shown to be a kind of regularization. Collins & Kohli applied $\ell 1$ regularization and shrinkage operators in the training process. However, it only reduced the weights by only $4\times$ with inferior accuracy. Tsuruoka et al. improved on this with $\ell 1$ regularization with superior compression, but the methods use SGD and has slow asymptotical convergence due to the inherent variance Johnson & Zhang (2013).

## 3 Mini-batch Non-convex SVRG

For Problem 1, a stochastic iterative learning algorithm estimate a stationary point $\mathbf{x}$ and achieve $\varepsilon$-accuracy in finite iterations satisfying $|| \bigtriangledown f(x)||^2 \leq \varepsilon$, which is termed of the $\varepsilon$-accurate solution. For a non-convex problem, the goal is to find a reasonable local minimum. However, the challenge is that gradients are easy to be stuck into saddle-point or a local minimum. As a result, such an algorithm aims to help gradients escape from saddle-point or local-minimal, e.g.(Ge et al., 2015) demonstrated that adding additional noise can help the algorithm escape from saddle points. To our best knowledge, there is no theoretically proof that can guarantee SVRG has faster rates of convergence than SGD. (Reddi et al., 2016) compared the Incremental First-order Oracle (IFO) complexity Agarwal & Bottou (2015) of SGD and SVRG on non-convex problem, $\mathcal{O}\left(1/\varepsilon^2\right)$ and $\mathcal{O}\left(n + (n^{\frac{2}{3}}/\varepsilon)\right)$ respectively. For

our analysis, whether non-convex SVRG can be efficiently close to reasonable optimal local minimum depends on the number of training samples. Suppose $f_i$ is non-convex for $i \in [n]$ and $f$ has $\varepsilon$-bounded gradients, the IFO complexity of mini-batch SGD with a adaptive learning rate is $\mathcal{O}\left(1/\varepsilon^2\right)$ and for mini-batch SVRG with a fixed learning rate $\mathcal{O}\left(n + (n^{\frac{2}{3}}/\varepsilon)\right)$. If the value of $\varepsilon$ is constant, the speed of convergence rates of SVRG depends on the number of training samples: when $n$ is small, SVRG is faster than SGD for non-convex optimization and vice versa. Our experiment results showed in Figure1 and Figure5can support our view.

### 3.1 Mini-batch Non-convex SVRG on Sparse Representation

In our case, SVRG is applied on sparse representation. However, if directly combining mini-batch non-convex SVRG with cumulative $\ell 1$ regularization (called *SVRG-cumulative-$\ell 1$*): let $u_k$ be the average value of the total $\ell 1$ penalty given by

$$u_k = \frac{\lambda}{M} \sum_{t=1}^{k} \eta_t. \tag{5}$$

At each training sample, weights that are used in current sample can be updated as

$$w_j^{k+\frac{1}{2}} = w_j^k - \eta_k \left( \frac{1}{B} \sum_{i=1}^{B} (\nabla \psi_i(w_j^k) - \nabla \psi_i(\tilde{w}_j)) + \tilde{\mu}_j \right) \tag{6}$$

$$\begin{aligned} &\textbf{if } w_j^{k+\frac{1}{2}} > 0 \textbf{ then} \\ &\quad w_j^{k+1} = \max(0, w_j^{k+\frac{1}{2}} - (u_k + q_j^{k-1})), \\ &\textbf{else if } w_j^{k+\frac{1}{2}} < 0 \textbf{ then} \\ &\quad w_j^{k+1} = \min(0, w_j^{k+\frac{1}{2}} + (u_k - q_j^{k-1})), \end{aligned} \tag{7}$$

where, $q_j^k$ is the total difference of two weights between the SGD update and the $\ell 1$ regularization update,

$$q_j^k = \sum_{t=1}^{k} (w_j^{t+1} - w_j^{t+\frac{1}{2}}), \tag{8}$$

where $t$ is an index to calculate cumulative value of $q$, the algorithm has two problems: (1) As we mentioned, SVRG on sparse representation cannot guarantee to be faster than SGD. Figure 1 shows that for small dataset (e.g. MNIST) the convergence of SVRG is faster than SGD but slower than SGD using a larger dataset (e.g. CIFAR-10), (2) The trade-off

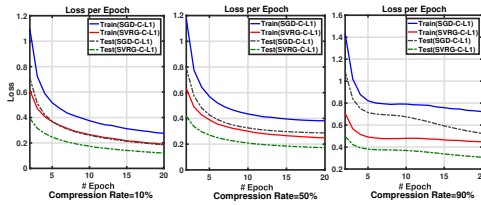
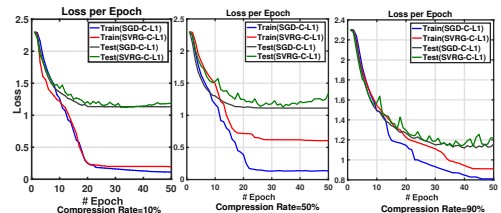

(a) MNIST dataset on LeNet-300-100 model     (b) CIFAR-10 dataset on LeNet-5 model

Figure 1: With cumulative $\ell 1$ regularization, we compare the convergence rates of SGD and SVRG. *SVRG-cumulative-$\ell 1$* has faster convergence rate in Figure 1(a). However, in Figure 1(b), *SGD-cumulative-$\ell 1$* can significantly converge into lower loss than *SVRG-cumulative-$\ell 1$* when compression rate equal 50% and 90%.

of *SVRG-cumulative-$\ell 1$* in the variance reduction versus the sparsity of the cumulative $\ell 1$ regularization. After the variance of the gradient is reduced by SVRG in Equation 6, the absolute value of the updated weight $w_j^{k+\frac{1}{2}}$ is higher than that using SGD, which causes SVRG

to have an adverse effect on the sparsity of $\ell 1$-regularization. Compared to ordinary SVRG, (Reddi et al., 2016) proposed an extension of SVRG: *MSVRG* that introduces adapts the learning rate, which guarantee that their method has equal or better than SGD. Therefore, similar to the method *MSVRG*, our method provides separate adaptive learning rates for *SVRG-cumulative-$\ell 1$*, which empirically demonstrates that it has faster convergence rate than SGD.

### 3.2 Delicate-SVRG-cumulative-$\ell 1$

To reduce the number of weights, we introduce our compression method *Delicate-SVRG-cumulative-$\ell 1$* that have two main improvements :

**(1) Separate Adaptive Learning Rate**   Learning rates play an important rule in effecting the convergence rate of optimization during the training process which must be chosen carefully to ensure that the convergence rate is fast, but not too aggressive in which case the algorithms may become unstable. Reddi et al. believe that adaptive learning rates can be applied with reduced variance to provide faster convergence rates on nonconvex optimization. As a result, the convergence rate of the algorithms can be improved if the learning rate is adaptively updated. Our algorithm includes three parameters to provide greater fidelity in controlling the convergence of gradients for implementation of the $\ell 1$ regularization.

Firstly, the learning rate $\gamma_k$ is chosen based on the learning rate from Collins et al. shown as,

$$\gamma_k = \frac{\eta_0}{1 + \pi(k/N)} \tag{9}$$

where $\eta_0$ is an initial learning rate with large value. Our experiments determined the parameters in three learning rates are over range of values, and a value of $\pi = 0.6$ as determined to be efficient. The learning rate schedule can emphasis the large distance between the gradient in the current optimization iteration and the sub-optimal solutions after every *m* iteration in the beginning, which avoids the current gradient being stuck in a local minimum at the start. It has a fast convergence rate to start with which decreases over time to minima local station.

The second learning rate, $\beta_k$, that reduces the variance of the *SVRG-cumulative-$\ell 1$* and better balances the trade-off in both of SVRG and cumulative $\ell 1$ regularization. $\beta_k$ is chosen such that $\beta_k > \gamma_k$ with slower convergence as

$$\beta_k = \frac{\eta_0}{1 + \alpha(k/N)^q}, \tag{10}$$

here $\beta_k = 0.75$, and the results of experiment is the best when $q = 3$ that can keep relatively large penalty of average gradients. During updating weight, it is efficient to prevent the absolute value of weight from being increased by SVRG, which can reduce the bad effect of $\ell 1$ regularization, and sparsity.

We retain the same learning rate $\eta_k$ for cumulative $\ell 1$ regularization Tsuruoka et al. (2009) shown as,

$$\eta_k = \eta_0 \alpha^{k/N} \tag{11}$$

The exponential decay ensures that the learning rates dose not drop too fast at the beginning and too slowly at the end.

**(2) Bias-based Pruning**   To further reduce the number of weights, we add a bias-based pruning $\tilde{b}$ after the $\ell 1$ regularization in each iteration. The pruning rule is based on following heuristic Fonseca & Fleming (1995): connections (weights) in each layer will be removed if their value is smaller than the network's minimal bias. If the absolute value of weight connections are smaller than the absolute value of the smallest bias of the entire network in each batch, these connections have least contribution to the node, which can be removed. In practice, bias-based pruning has no effect on train and test loss.

Consequently, *Delicate-SVRG-cumulative-ℓ1* that incorporates the adaptive learning rate schedules and bias-based pruning as,

$$
w_j^{k+\frac{1}{2}} = w_j^k - \left( \frac{\gamma_k}{N} \sum_{i=1}^{N} (\nabla \psi_i(w_j^k) - \nabla \psi_i(\tilde{w}_j)) + \beta_k \tilde{\mu}_j \right)
$$

$$
\textbf{if } w_j^{k+\frac{1}{2}} > 0 \textbf{ then}
$$

$$
w_j^{k+1} = \max(0, w_j^{k+\frac{1}{2}} - (u_k + q_j^{k-1} + \tilde{b})),
$$

$$
\textbf{else if } w_j^{k+\frac{1}{2}} < 0 \textbf{ then}
$$

$$
w_j^{k+1} = \min(0, w_j^{k+\frac{1}{2}} + (u_k - q_j^{k-1} - \tilde{b})).
$$

(12)

The pseudo code of our method is illustrated as Algorithm 1 in the Appendix.

## 4 Weight Quantization for Bit-depth Reduction

To further compress the model, weight quantization can significantly reduce memory requirement by reducing bit precision. We quantize to 3-bit after reducing by *Delicate-SVRG-cumulative-ℓ1* for convolutional layers and encode 5 bits for fully connected layers. Consequently, we propose our final compression method *Weight Reduction Quantisation*.

Table 1: Comparison of the compression results of the pruning method from Han et al. (2016) and our method in each layer. Using MNIST dataset train and test on LeNet-300-100 1(a) and LeNet-5 model1(b). D is *Delicate-SVRG-cumulative-ℓ1* and Q is weight quantization.

(a) MNIST dataset with LeNet-300-100 model.

| Layer | Original network | #Weights (D) | Memory (D+Q) | Compress rate (D) | Compress rate (D+Q) | Deep compression Han et al. (2016) Compress rate |
|---|---|---|---|---|---|---|
| ip1 | 235K(940KB) | 8.0K | 14.36KB | 3% | 1.63% | 2.32% |
| ip2 | 30K(120KB) | 2.5K | 3.392KB | 8.3% | 2.82% | 3.04% |
| ip3 | 1K(4KB) | 0.3K | 0.308KB | 30% | 7.7% | 12.70% |
| Total | 266K(1070KB) | 10.8K | 18.06KB | 4%(25×) | 1.68%(**60×**) | 2.49%(**40×**) |
| Top-1 Error | 1.64% | - | - | 1.58% | **1.57%** | 1.58% |

(b) MNIST dataset with LeNet-5 model.

| Layer | Original network | D-SVRG-C-L1 (D) | Memory (D+Q) | Compress rate (D) | Compress rate (D+Q) | Deep compression Han et al. (2016) Compress rate |
|---|---|---|---|---|---|---|
| conv1 | 0.5K(2KB) | 0.33K | 1.16KB | 78% | 58% | 67.85% |
| conv2 | 25K(100KB) | 3K | 2.42KB | 12% | 2.42% | 5.28% |
| ip1 | 400K(1600KB) | 32K | 24KB | 3.7% | 1.5% | 2.45% |
| ip2 | 5K(40KB) | 0.95K | 2.112KB | 17% | 5.28% | 6.13% |
| Total | 431K(1720KB) | 35K | 30KB | 4.5%(22×) | 1.8%(**57×**) | 2.55%(**39×**) |
| Top-1 Error | 0.80% | - | - | 0.74% | **0.737%** | 0.74% |

## 5 Experiments

In order to estimate and compare the effect of our compression method on different topologies, e.g. fully connected networks and convolutional networks, we select deep neural networks (DNNs) and convolutional neural networks (CNNs). The DNN chosen is LeNet-300-100 which has two fully connected layers as hidden layers with 300 and 100 neurons respectively. The CNN chosen is LeNet-5 which has two convolutional layers and two fully connected layers. We evaluate the performance of our new compression method using MNIST,

and CIFAR as benchmarks. MNIST (LeCun et al., 2001) is a set of handwritten digits which is a commonly used dataset in machine learning. It has 60,000 training examples and 10,000 test samples. Each image is grey-scale with $28 \times 28$ pixels. CIFAR-10 is a dataset that has 10 classes with 5,000 training images and 1,000 test images in each class. In total, it contains 50,000 training images and 10,000 test images with $32 \times 32$ pixels. CIFAR-10 images are RGB. Two types of error rate are used to measure the performance of models, which are top-1 and top-5 error rate. Here, we consider top-1 error on MNIST, while top-5 error on CIFAR-10 because many images in CIFAR are small and ambiguous. Our compression method was implemented using Caffe[1].

## 5.1 Comparison with leading results

Applying *Weight Reduction Quantisation* to the MNIST dataset, we choose the results with the best combination of compression and error rate for comparison. Our method can reduce 98% of the memory requirements with a 1.57% test error rate on the LeNet-300-100 model and 98% of the parameters with a 0.74% test error on the LeNet-5 model. In Table 1, the compression pipeline is summarised with weight statistics in comparison to the method from Han et al. (2016). In our first stage *Delicate-SVRG-cumulative-ℓ1* that focus on reducing the number of weights, we compare the results of pruning method from Han et al. (2016) that is the first stage of their compression method. The two tables show that both *Delicate-SVRG-cumulative-ℓ1* and Han et al. pruning method can significantly remove many weights in the fully connected layers. For LeNet-300-100 models, the number of weights in the first fully connected layers (ip1) contains about 88% of the total number of weights and this can be compressed by 97% by *Delicate-SVRG-cumulative-ℓ1*. Furthermore, both *Delicate-SVRG-cumulative-ℓ1* and pruning method have very similar compression rate in convolutional layers (conv1 and conv2) in reducing the number of weights in LeNet-5 model, but *Delicate-SVRG-cumulative-ℓ1* is more effective to reduce the number of weights of the two fully connected layers (ip1 and ip2) sparse. Both *Delicate-SVRG-cumulative-ℓ1* and Han et al. pruning method can achieve lower test error than that of uncompressed models, whilst delivering overall compression rates up to $25\times$ and $12\times$ respectively. The

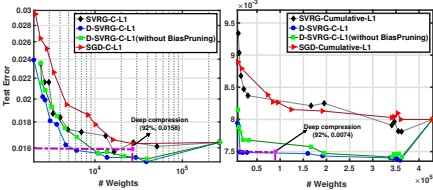 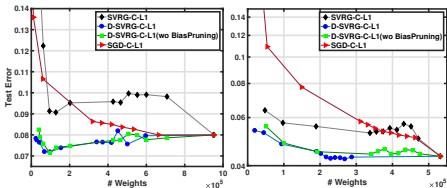

(a) MNIST dataset on LeNet-300-100 model (left) and LeNet-5 (right): error rate is top-1 error

(b) CIFAR-10 dataset on LeNet-300-100 model (left) and LeNet-5 (right): error rate is top-5 error

Figure 2: Four $\ell 1$ regularization compression methods experiment on two deep learning models, including LeNet-300-100 and LeNet-5 using MNIST datasets and CIFAR-10.

second stage is to further compress the model by bit-depth reduction, and Table1 shows our method *Weight Reduction Quantisation* that combines *Delicate-SVRG-cumulative-ℓ1* with bit-depth reduction can achieve 1.56% error rate on MNIST, and 0.737% error rate on CIFAR-10, where the two errors are all lower than that of original model. The compression rates are up to $60\times$ in LeNet-300-100 and up to $57\times$ in LeNet-5 model.

---

[1]Caffe is a deep learning framework. Source code can be download: http://caffe.berkeleyvision.org

## 5.2 Evaluation the Trade-off Between Memory Requirements and Performance

Focusing on *Delicate-SVRG-cumulative-ℓ1* to examine the performance of method at different compression rates controlled by threshold λ, we compare the performance of different model-compression based on ℓ1 regularization over the range of memory requirements. Figure 2 shows how the test error rate and weight sparsity vary as the regularization parameter λ is adjusted. Where pareto fronts are not available for comparison, we compare with a single trade-off and determine the related performance by the side of the pareto front that the point lies.

**LeNet on MNIST** Figure 2(a) shows LeNet on MNIST. Compared with *SVRG-cumulative-ℓ1*, *SGD-cumulative-ℓ1* has the better ability of compression, but the error rate is higher due to the variance generated by SGD optimiser. Replacing SGD with SVRG, *SVRG-cumulative-ℓ1* reduces the test error but the compression ability is also reduced. The *Delicate-SVRG-cumulative-ℓ1* method has the least number of weights and the best performance having the lowest test error for almost every compression value. Its performance is similar with the method without bias-based pruning, which means that adding bias-based pruning can further reduce the number of weights without side-effect on the performance. The pink box on 2(a) showed that the results within the box is better than pink point.

**LeNet on CIFAR-10** Figure 2(b) shows LeNet on CIFAR-10 dataset that is a larger and more complicated dataset than MNIST. *SVRG-cumulative-ℓ1* has chances to achieve lower test error than *SGD-cumulative ℓ1* but can not guarantee that the performance is always better than *SGD-cumulative-ℓ1*. *Delicate-SVRG-cumulative-ℓ1* method has better performance than the other methods. Its performance is further enhanced by adding bias-based pruning. Consequently,*Delicate-SVRG-cumulative-ℓ1* can be effectively applied in LeNet-300-100 and LeNet-5 models without accuracy loss when applied to MNIST and CIFAR-10.

## 5.3 Combining Delicate-SVRG-cumulative-ℓ1 and Weight Quantization

Figure3 shows the test error at different compression rates for *Delicate-SVRG-cumulative-ℓ1* and weight Quantization. Individually, weight quantization can reduce more memory before the test error increases significantly in *Delicate-SVRG-cumulative-ℓ1* using MNIST dataset, but the reverse results applied on CIFAR-10 dataset. However, if combining together, the approach consistently outperforms.

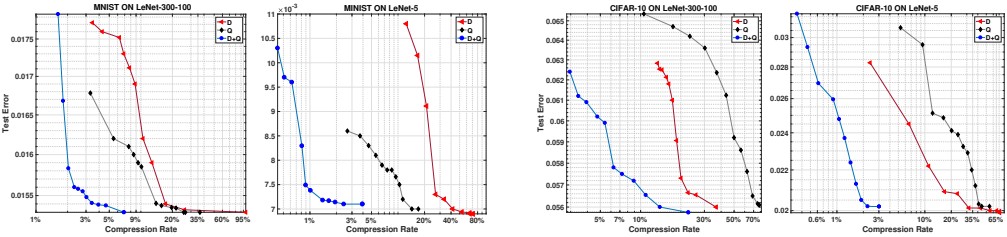

(a) MNIST dataset on LeNet-300-100 model (left) and LeNet-5 (right).

(b) CIFAR-10 dataset on LeNet-300-100 model (left) and LeNet-5 (right).

Figure 3: The test error with compression rate under different compression methods. D is Delicate-SVRG-cumulative-ℓ1, Q is weight quantization. Combining Delicate-SVRG-cumulative-ℓ1 with weight quantization can achieve the best performance.

### 5.4 COMPARISON OF CONVERGENCE RATES

To confirm the theoretical insights that our method has no bad effect on convergence rate to achieve similar fast convergence with *SGD-cumulative-ℓ1* or *SVRG-cumulative-ℓ1*, we calculate the training loss of two LeNet models on MNIST and CIFAR datasets during increasing iterations. In Figure 4(a), all methods have similar convergence rates in LeNet-300-100. In all of our experiments, *Delicate-SVRG-cumulative-ℓ1* has same or lower training loss and faster convergence rate than other methods, meaning that adaptive learning rate can help SVRG with cumulative ℓ1 regularization to escape the local minimum in the beginning and quickly converge to a good local minimum within finite training iterations. Moreover, *Delicate-SVRG-cumulative-ℓ1* without bias-based pruning has a similar train loss, which illustrates that adding bias-based pruning in ℓ1 regularization has no obvious bad effect on the convergence of weights. Consequently, applying adaptive learning rates, *Delicate-SVRG-cumulative-ℓ1* is a efficient compression method for neural network problems.

## 6 DISCUSSION

In this paper, we proposed *Weight Reduction Quantisation* that efficiently compressed neural networks without scarifying accuracy. Our method has two stages that reduce the number of weights and reduce the number of bits to store each weight. We show that SVRG and cumulative ℓ1 regularization can improve over SGD and ℓ1-regularization. By combining them, we have presented a new compression method *Delicate-SVRG-cumulative-ℓ1* that can efficiently reduce the number of parameters by the separate adaptive learning rates. The three adaptive learning rates are applied on SVRG and cumulative ℓ1 penalty, which provides a high accuracy and reduced number of weights. Besides, our method improved SVRG that can be used on non-convex problem with fast convergence rate. In our experiments on LeNet-300-100 and LeNet-5, our method can significantly reduce the memory requirements up to $60\times$ without accuracy loss. After compression by our method, a compact deep neural network can be efficiently deployed on an embedded device with performance of the original model.

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

---

**Algorithm 1** Delicate-SVRG-cumulative-$\ell$1: Stochastic descent training with cumulative $\ell$1 penalty

---

**procedure** TRAIN( $\lambda$)
    $u \leftarrow 0$
    $\tilde{\mu} \leftarrow 0$
    Initial $w_j$ and $q_j$ with zero for all number of weights M
    **for** k=0 to Maximal Iterations **do**
        $\gamma \leftarrow \dfrac{\eta_0}{1 + \pi(k/N)}$
        $\beta \leftarrow \dfrac{\eta_0}{1 + \alpha(k/N)^3}$
        **for** t=0 to k **do**
            $\eta \leftarrow \eta_0 \alpha^{t/N}$
        **end for**
        $u \leftarrow u + \eta\lambda/M$
    **end for**
    **for** $j \in$ features used in sample i **do**
        randomly select m features from train samples
        $w_j \leftarrow w_j - \left(\frac{\gamma_k}{N}\sum_{i=1}^{N}(\nabla\psi_i(w_j) - \nabla\psi_i(\tilde{w}_j)) + \beta_k\tilde{\mu}\right)$
        $\nabla\psi_i(\tilde{w}_j) = \nabla\psi_i(w_j)$
        **if** $w_j$ and $\tilde{w}_j$ converge to the same weights **then**
            $\tilde{\mu} = 0$
        **end if**
        $\tilde{\mu} \leftarrow \tilde{\mu} + \frac{1}{N}\nabla\psi_i(\tilde{w})$
    **end for**
**end procedure**
**procedure** APPLY PENALTY(i)
    $z \leftarrow w_j$
    $\tilde{b}$ is minimal bias in all layers.
    **if** $w_j > 0$ **then**
        $w_j \in \max(0, w_j - (u + q_j + \tilde{b}))$,
    **else**
        **if** $w_j < 0$ **then**
            $w_j \in \min(0, w_j + (u - q_j - \tilde{b}))$,
        **end if**
    **end if**
    $q_j \leftarrow q_j + (w_j - z)$
**end procedure**

---

# 7 APPENDIX

## 7.1 THE ALGORITHM OF DELICATE-SVRG-CUMULATIVE-$\ell$1

## 7.2 COMPARISON OF THE CONVERGENCE RATES OF BETWEEN OUR METHOD AND SVRG AND SGD WHEN COMBINING WITH $\ell$1 REGULARIZATION.

The results showed in Figure4.

## 7.3 USING MULTIPLE INITIALIZATIONS TO COMPARE THE PERFORMANCE OF OUR METHOD AND OTHER THREE METHODS.

The experiments were run with multiple initializations and there was some small variability in the results. However, the relative performance of the our method is always better than SVRG and SGD combining with cumulative $\ell$1 regularization. The results showed in Figure5

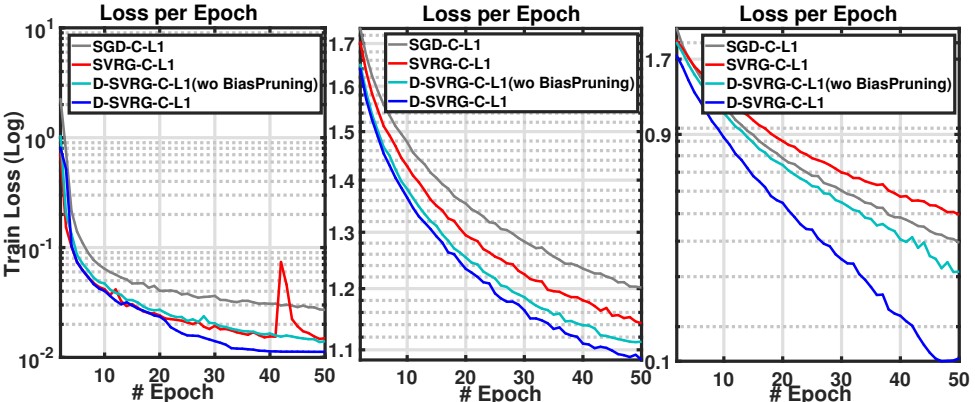

(a) The train loss: MNIST dataset on LeNet-5 (left) and CIFAR-10 dataset on LeNet-300-100 (middle) and LeNet-5 (right)

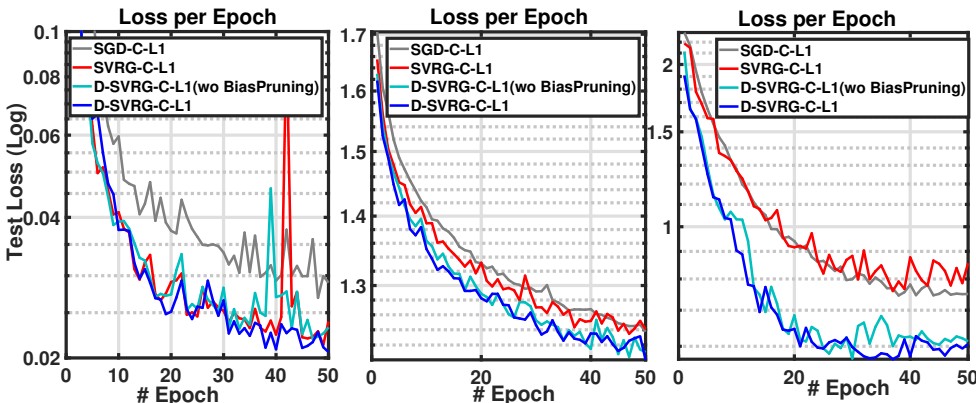

(b) The test loss: MNIST dataset on LeNet-5 (left) and CIFAR-10 dataset on LeNet-300-100 (middle) and LeNet-5 (right)

Figure 4: Estimate the convergence rate when using four compression methods, including our method *Delicate-SVRG-cumulative-ℓ1*, *Delicate-SVRG-cumulative-ℓ1 (without BiasPruning)* that without bias-based pruning in ℓ1 regularization,*SVRG-cumulative-ℓ1* and *SGD-cumulative-ℓ1*, on LeNet-300-100 and LeNet-5 models with MNIST and CIFAR-10 datasets. Here we choose the compression rate that equal 90% to observe training and test loss. For MNIST dataset, we did not notice subtle difference train and test loss on LeNet-300-100 model generated by four methods.

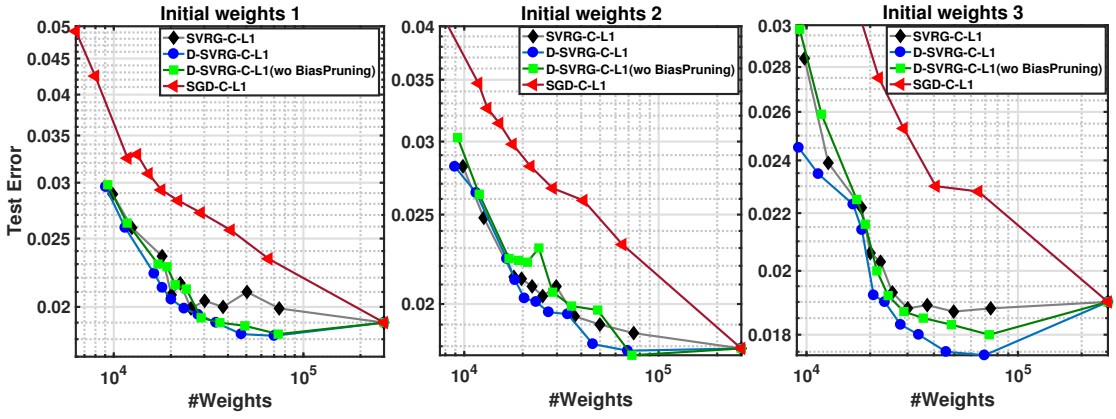

(a) MNIST dataset on LeNet-300-100

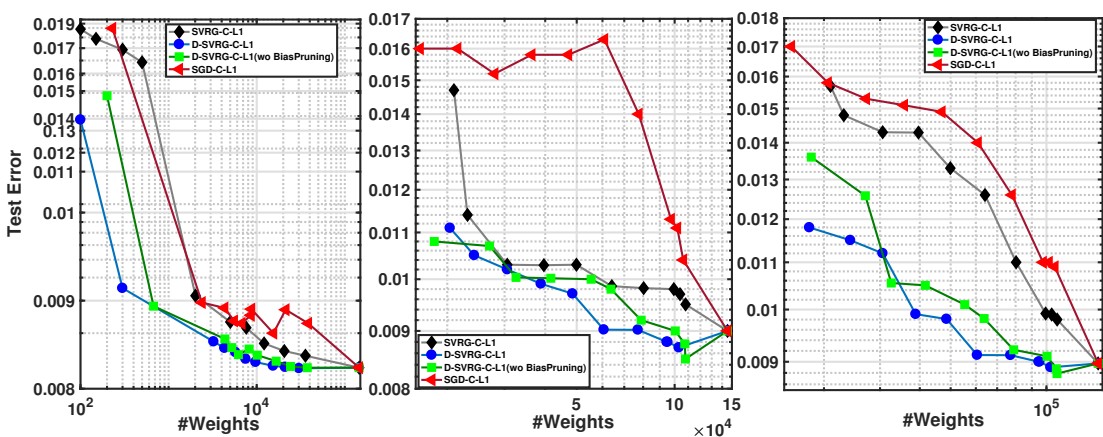

(b) MNIST dataset on LeNet-5

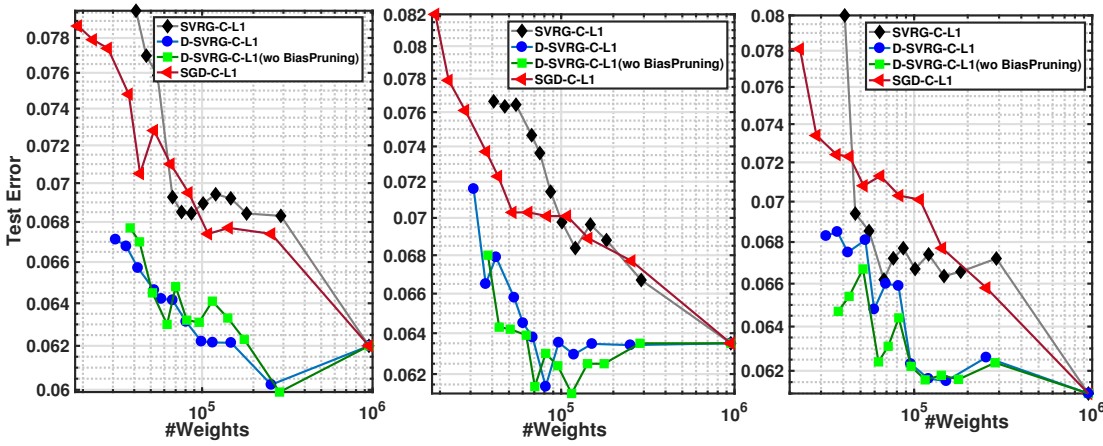

(c) CIFAR-10 dataset on LeNet-300-100

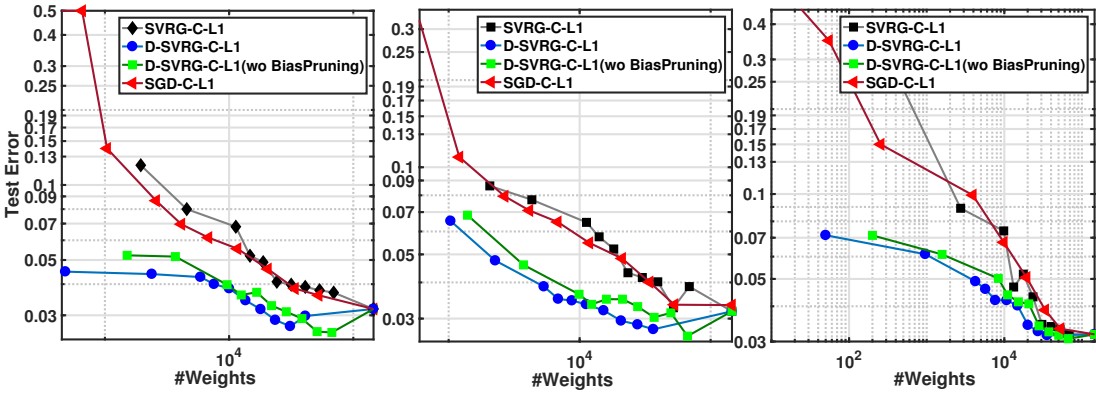

(d) CIFAR-10 dataset on LeNet-5

Figure 5: Using three types of initial weights, we compare our method with other three methods. D-SVRG-C-L1 and D-SVRG-C-L1(wo BiasPruning) are always better than other two methods. This experiment also can verify the our view that the performance of SVRG is better or worse than SGD that depends on the number of training samples. In our experiment, if choosing small dataset (e.g. MNIST), SVRG is better than SGD. Otherwise, if choosing relatively large dataset (e.g. CIFAR-10), SVRG is worse than SGD.

