# OpenReview forum: "Sparse Regularized Deep Neural Networks For Efficient Embedded Learning"
_ICLR.cc/2018/Conference — Reject_

### Official Review · AnonReviewer3 · 2017-11-25
**Sparse Regularized Deep Neural Networks For Efficient Embedded Learning**

**Rating:** 4
**Confidence:** 4

**Review:**

Summary:
Paper proposes the compression method Delicate-SVRG-cumulative-L1 (combining minibatch SVRG with cumulative L1 regularization) which can significantly reduce the number of weights without affecting the test accuracy. Paper provides numerical experiments for MNIST and CIRAR10 on LeNet-300-100 and LeNet-5.

Comments:
Up to my knowledge, Han et. al (2016) is not the leading result. There are (at least) two more results which are better than Han et. al. (2016) and also better than your results for LeNet-300-100 and LeNet-5 (MNIST), which were already published at ICML 2017 and NIPS 2016:
http://papers.nips.cc/paper/6165-dynamic-network-surgery-for-efficient-dnns.pdf
http://proceedings.mlr.press/v70/molchanov17a/molchanov17a.pdf

There is no theory supporting the proposed method (which is the combination of some existing methods). Therefore, you should provide more experiments to show the efficiency. MNIST and CIFAR10 on LeNet-300-100 and LeNet-5 are quite standard that people have already shown.

Moreover, there is no guarantee for sparsity by using L1 regularization on nonconvex problems.

Minor comments:
Page 3, section 2, first paragraph: typo in the last sentence: “dose” -> “does”
Same typo above for page 5, the sentence right before (2) Bias-based pruning

---

> ### Author Response · Authors · 2018-01-05
> **Compared with other two methods**
>
> 1: Up to my knowledge, Han et. al (2016) is not the leading result. There are (at least) two more results which are better than Han et. al. (2016) and also better than your results for LeNet-300-100 and LeNet-5 (MNIST), which were already published at ICML 2017 and NIPS 2016.
> [1] http://papers.nips.cc/paper/6165-dynamic-network-surgery-for-efficient-dnns.pdf
> [2] http://proceedings.mlr.press/v70/molchanov17a/molchanov17a.pdf
>
> Our method can achieve lower test error than [1] and [2] in both LeNet-300-100 and LeNet-5 model. If we keep the same error with [1] and [2], the compression rate of our method is shown below in two tables, which showed that our method is competitive with other methods.
>
> Model          Params.% Method[1]      Params.%(Ours)           Test error
> LeNet-5                    0.9%                               0.34%                             0.91%
> LeNet-300-100        1.8%                               0.78%                             2.28%
> Model          Params.% Method[2]      Params.%(Ours)           Test error
> LeNet-5                    0.36%                               2%                                0.75%
> LeNet-300-100        1.4%                               0.97%                             1.92%

---

### Official Review · AnonReviewer1 · 2017-11-28
**The authors use l-1 regularized SVRG to promotes sparsity in the trained model. However, the paper lacks comparisons with some key literature, and experimentally the benefit of SVRG over SGD does not seem substantial.**

**Rating:** 4
**Confidence:** 5

**Review:**

The authors present an l-1 regularized SVRG based training algorithm that is able to force many weights of the network to be 0, hence leading to good compression of the model.  The motivation for l-1 regularization is clear as it promotes sparse models, which lead to lower storage overheads during inference. The use of SVRG is motivated by the fact that it can, in some cases, provide faster convergence than SGD.

Unfortunately, the authors do not compare with some key literature. For example there has been several techniques that use sparsity, and group sparsity [1,2,3], that lead to the same conclusion as the paper here: models can be significantly sparsified while not affecting the test accuracy of the trained model.

Then, the novelty of the technique presented is also unclear, as essentially the algorithm is simply SVRG with l1 regularization and then some quantization. The experimental evaluation does not strongly support the thesis that the presented algorithm is much better than SGD with l1 regularization. In the presented experiments, the gap between the performance of SGD and SVRG is small (especially in terms of test error), and overall the savings in terms of the number of weights is similar to Deep compression. Hence, it is unclear how the use of SVRG over SGD improves things. Eg in figure 2 the differences in top-1 error of SGD and SVRG, for the same number of weights is very similar (it’s unclear also why Fig 2a uses top-1 and Fig 2b uses top-5 error). I also want to note that all experiments were run on LeNet, and not on state of the art models (eg ResNets).

Finally, the paper is riddled with typos. I attach below some of the ones I found in pages 1 and 2

Overall, although the topic is very interesting, the contribution of this paper is limited, and it is unclear how it compares with other similar techniques that use group sparsity regularization, and whether SVRG offers any significant advantages over l1-SGD.

typos:
“ This work addresses the problem by proposing methods Weight Reduction Quantisation”
-> This work addresses the problem by proposing a Weight Reduction Quantisation method

“Beside, applying with sparsity-inducing regularization”
-> Beside, applying sparsity-inducing regularization

“Our method that minibatch SVRG with l-1 regularization on non-convex problem”
-> Our minibatch SVRG with l-1 regularization method on non-convex problem

“As well as providing,l1 regularization is a powerful compression techniques to penalize some weights to be zero”
-> “l1 regularization is a powerful compression technique that forces some weights to be zero”

 The problem 1 can
->  The problem in Eq.(1) can

“it inefficiently encourages weight”
-> “it inefficiently encourages weights”

————

[1] Learning Structured Sparsity in Deep Neural Networks
http://papers.nips.cc/paper/6504-learning-structured-sparsity-in-deep-neural-networks.pdf

[2] Fast ConvNets Using Group-wise Brain Damage
https://arxiv.org/pdf/1506.02515.pdf

[3] Sparse Convolutional Neural Networks
https://www.cv-foundation.org/openaccess/content_cvpr_2015/papers/Liu_Sparse_Convolutional_Neural_2015_CVPR_paper.pdf

---

> ### Author Response · Authors · 2018-01-05
> **Compared with reference [1] [2] and [3]**
>
> [1] and [2] achieved test errors on MNIST dataset with a LeNet network of 1% and 1.71% respectively and these are higher than our method. In [1], the remaining weights were about 2.625K. If we keep the same test error of 1%, our method can reduce this to about 0.5K as shown in Figure 4b. [2] do not provide the number of weights after compression by L1 regularization in the experiment on MNIST dataset in LeNet model. [3] do not provide the experiment on MNIST dataset. Hence, we cannot directly compare with their methods. So far our experiments use two datasets and two different models (LeNet-300-100 and LeNet-5). We aim to show the performance of our method on dense-based models and convolutional-based models. In our future work, we will do more experiments on different datasets and models (e.g. CIFAR-100 and ImageNet datasets, and AlexNets , VGG and ResNets models. )

---

### Official Review · AnonReviewer2 · 2017-11-30
**Confusing**

**Rating:** 2
**Confidence:** 3

**Review:**

It is very hard to follow this work, it feels like it tries to get several messages across while none of them properly. The work further contains number of unclear or incorrect claims, meaningless comparison with existing work, and unbelievable results ("0.737% error rate" on CIFAR-10).

In introduction, first, the paper seems to be about L1-regularization, with few motivating remarks valid only for convex problems, then about novel optimization method, and suddenly main contribution is reducing memory requirements. Further, part on "Cumulative l1 regularization" need to be better explained if, as it seems, plays important role in what you do. In discussion about SVRG, I don't understand how claims about convergence and batch size make sense, please provide reference, and how is it important for what you do later. When you say "Hence, a promising approach is to use..." I don't understand how it either follows from discussion above, nor what is the problem that you address.
In Main Contributions, 2.1 - "we analyse non-convex SVRG" - I don't see any kind of analysis in the paper.

Sec 3. you use IFO of Agarwal and Bottou which is known not to include this kind of algorithm - see large red box above abstract in the last version of the cited paper. Even then it is not clear what you try to say in the section, and whether any of it is new.

Sec 3.1. What is the notion of "larger dataset"? You regard CIFAR-10 as larger than MNIST.

Sec 4. After 4 pages of discussion on optimization algorithms, you write (very ambiguous) 4 lines about quantization, and compare against work not related to optimization at all. No explanation of what is presented in the table nor notation used. It requires lot of guessing to see what you try to do.
If I guessed correctly, you propose optimization method used together with particular objective function to train a model that is sparse in its final trained form, and then reduce numerical precision used to represent the model. And compare that to Han et al.
1. If this is what you try to do, it is never clearly stated it up to this point, and much of the preceding text is irrelevant and it is sufficient to just refer to existing work... I now see you have a similar statement in Discussion, but if this is what you try to do and has to be explained at the beginning.
2. It does not make any sense to compare against Han et al (precisely against the numbers presented in their paper), as you are compressing something else. If applied to your trained model, I believe it would achieve significantly better result.

I did not properly look at the experiments, as it is not clear what you do/propose in first place, and you seems to report 0.737% error rate on CIFAR-10, and in the appendix, plots for CIFAR-10 show convergence to ~3% test error with LeNet-5.

---

> ### Author Response · Authors · 2018-01-05
> **Explain our main work and objective.**
>
> Thank you for your reading. I'll reply your several questions as below：
> 1: There is a typo in here. The 0.737% error rate refers to the MNIST dataset using the LeNet-5 model.
> 2: The main concern of our work is to reduce the memory requirements of the neural network. L1 regularization is one compression technique that is efficient in reducing the number of parameters whilst maintaining accuracy. SVRG is better than SGD at efficiently finding the solution in strongly convex problems. However, using SVRG with L1 regularization (SVRG-C-L1) is not efficient when applied in non-convex problems such as neural networks. As a result, our work aims to improve this situation. We have modified SVRG-C-L1 by using adaptive learning rates, with the results showing that our method is better suited in non-convex problem. In our main contribution, we analyze and provide the condition when SVRG has faster convergence rate than SGD in section 3 “mini-batch non-convex SVRG” and sub section 3.1 “Mini-batch Non-convex SVRG on Sparse Representation” using training loss as a way to measure the convergence rates. (https://papers.nips.cc/paper/4937-accelerating-stochastic-gradient-descent-using-predictive-variance-reduction.pdf
>
> 3: Here, IFO is one type of complexity proposed by Agarwal and Bottou (2015).  http://proceedings.mlr.press/v37/agarwal15.pdf
> In this section, we followed the work from Reddi et.at 2016 that compared the IFO complexity of different algorithms (such as SGD and SVRG). We determined that SVRG has better performance of optimization than SGD (in other words, SVRG has faster speed of convergence than SGD) in non-convex problems, but this depends on the number of training samples. In our modified method, we experimented with two datasets and two models and showed that our method has the fastest speed of convergence than SVRG and SGD in figure 4.
>
> 3.1: CIFAR-10 has 163MB and MNIST has about 3MB. MNIST images are smaller (1,28,28) than the CIFAR-10 (3,224,224).
>
> 4: Table I explains the details in section 5.1 and the notation is explained in the table caption. D is our method that reduces the number of weights and Q is weight quantization that reduces the bit precision for storing each weight. D+Q represents both steps of weight reduction and quantization.
>
> 4.1 and 4.2: Memory reduction is our main objective. So we first use the same model and datasets to compare the compression rate of our method with the methods of others.  Secondly, we compared our results with other related L1 regularization compression techniques that use different optimization methods (SGD and SVRG), and show our method has faster convergence rates than other optimizations on different size of datasets.

---

### Decision · Program_Chairs · 2018-01-29
**ICLR 2018 Conference Acceptance Decision**

**Decision:**

Reject

**Comment:**

Dear authors,

I agree with the reviewers that the paper tries to do several things at once and the results are not that convincing. Overall, this work is mostly incremental, which is fine if there is no issue in the execution. Thus, I regret to inform you that this paper will not be accepted to ICLR.